🔓 | **Open Peer Review** | *Clinical Microbiology* | *Research Article*

# Identifying the irrationality of the diagnosis of "pertussis-like syndrome" to enhance diagnostic accuracy

Wei Shi,[1] Yahong Hu,[1] Qinghong Meng,[1] Guoshuang Feng,[2] Xinyu Wang,[2] Kaihu Yao[1]

**ABSTRACT** The purpose of this study is to assess the rationale and epidemiological patterns of "pertussis-like syndrome" diagnoses. A comprehensive analysis of demographic, epidemiological, and etiological characteristics was conducted on 10,561 diagnosed "pertussis-like syndrome" cases across 33 Chinese hospitals. Post-coronavirus disease 2019 pandemic, the incidence of "pertussis-like syndrome" increased significantly. Infants under 1 year old accounted for 69.73% of these cases, and severe outcomes were particularly prevalent among younger infants. Among those admitted to the intensive care unit, 83.03% were under 6 months of age, and 75.00% of the four reported deaths occurring in infants were younger than 3 months. While infants under 1 year consistently represented over half of annual cases, their proportion declined from 82.93% in 2016 to 51.21% in 2022. In contrast, there has been a notable rise in cases among children older than 3 years. It is important to highlight that only 4.37% of cases were exclusively diagnosed as "pertussis-like syndrome," with the majority of patients presenting with comorbidities, particularly lower respiratory tract infections (93.61%). The common pathogens identified in the records included respiratory syncytial virus, *Mycoplasma pneumoniae*, *Haemophilus influenzae*, and parainfluenza virus. "Pertussis-like syndrome" exhibits a high degree of overlap with pertussis in terms of the age distribution of susceptible populations and epidemiological patterns. To improve diagnostic accuracy, we recommend strengthening laboratory testing in suspected "pertussis-like syndrome" cases to confirm or rule out pertussis. For cases with identified pathogens that are not *Bordetella pertussis*, a precise pathogen-specific diagnosis should be established rather than relying on the ambiguous label of "pertussis-like syndrome."

**IMPORTANCE** This study highlights the critical importance of reevaluating the diagnosis of "pertussis-like syndrome" to improve diagnostic accuracy and patient outcomes. The global resurgence of pertussis has underscored the need for precise identification of respiratory infections, particularly in pediatric populations. Our analysis of 10,561 cases across 33 hospitals in China revealed significant overlaps between "pertussis-like syndrome" and pertussis in terms of age distribution and epidemiological patterns. Cases diagnosed as "pertussis-like syndrome" may include undetected cases of pertussis. Moreover, the broad, ambiguous label of "pertussis-like syndrome" often masks the true causative pathogens. This imprecise diagnosis hinders targeted treatment and public health surveillance. Given advancements in pathogen detection technologies, we advocate for abandoning the "pertussis-like syndrome" label in favor of precise, pathogen-specific diagnoses. This shift may enhance diagnostic clarity, optimize clinical management, and strengthen efforts to monitor and control respiratory infections globally.

**KEYWORDS** pertussis-like syndrome, pertussis, diagnosis, etiological test

**Peer Reviewer** Jikui Deng, Shenzhen Children's Hospital, Shenzhen, China

Address correspondence to Xinyu Wang, wangxy_bch@163.com, or Kaihu Yao, yaokaihu@bch.com.cn.

The authors declare no conflict of interest.

The global resurgence of pertussis has emerged as a significant public health challenge in recent years. Since early 2023, the substantial rise in reported pertussis cases across multiple countries (1–3) has underscored the urgent need for improved detection, diagnostic protocols, and surveillance systems. In our epidemiological analysis of hospitalized pertussis cases in China, we identified a considerable proportion of patients diagnosed with "pertussis-like syndrome."

"Pertussis-like syndrome" is clinically defined as a constellation of respiratory symptoms resembling those of pertussis but caused primarily by pathogens other than *Bordetella pertussis* (4). Affected children typically present with paroxysmal spasmodic coughing, facial flushing, and a characteristic high-pitched inspiratory "whooping" sound, which are hallmark features of classic pertussis. The etiological agents commonly associated with this syndrome include respiratory syncytial virus, adenovirus, *Streptococcus pneumoniae*, *Haemophilus influenzae*, and *Mycoplasma pneumoniae*, among others (5, 6). However, in clinical practice, the diagnosis of "pertussis-like syndrome" is frequently established without prior laboratory confirmation to rule out *Bordetella pertussis* infection. Notably, the diagnosis of "pertussis-like syndrome" is no longer classified under the current International Classification of Diseases (ICD) (7).

With the advancement of pathogen detection technologies, the identification rates of these respiratory pathogens have significantly improved. This progress, however, raises critical questions. Are the current diagnostic criteria for "pertussis-like syndrome" sufficiently accurate? What are the distinctive epidemiological and clinical characteristics of these cases? Are the implicated pathogens being reliably detected? And, crucially, could these cases be better categorized within the existing ICD framework?

To address these questions, we conducted a comprehensive analysis of demographic and clinical data from pediatric patients diagnosed with "pertussis-like syndrome" across 33 hospitals in China. Our objectives were to clarify the clinical and etiological profiles of these cases and to improve diagnostic accuracy.

## MATERIALS AND METHODS

### Data source

This study conducted a retrospective review and analysis of discharge data for cases diagnosed with "pertussis-like syndrome" from the Futang Research Center of Pediatric Development (FRCPD), a leading non-profit organization dedicated to pediatric medical research in China. Established on 23 August 2016, with authorization from the Ministry of Civil Affairs of the People's Republic of China, FRCPD comprises 69 provincial and municipal member hospitals. Additional details about FRCPD can be accessed through their official website (8).

The data for this study were collected from 11 provincial and 22 municipal hospitals spanning 25 provinces across all seven regions of China, from January 2016 to December 2022. The data were retrieved from the FUTang Updating Medical Records Database, and further specifics regarding data collection and cleaning methodologies are outlined in a prior publication (9). The data analyzed in this study were exclusively derived from hospitalized patients and did not include those who visited the emergency department but were not subsequently admitted.

### Inclusion criteria

This study incorporated cases identified as "pertussis-like syndrome" while excluding those diagnosed with parapertussis or confirmed pertussis.

### Variables and outcomes

Detailed sociodemographic data, including age, gender, ethnicity, and place of residence, as well as illness-related factors such as etiology, complications, comorbid

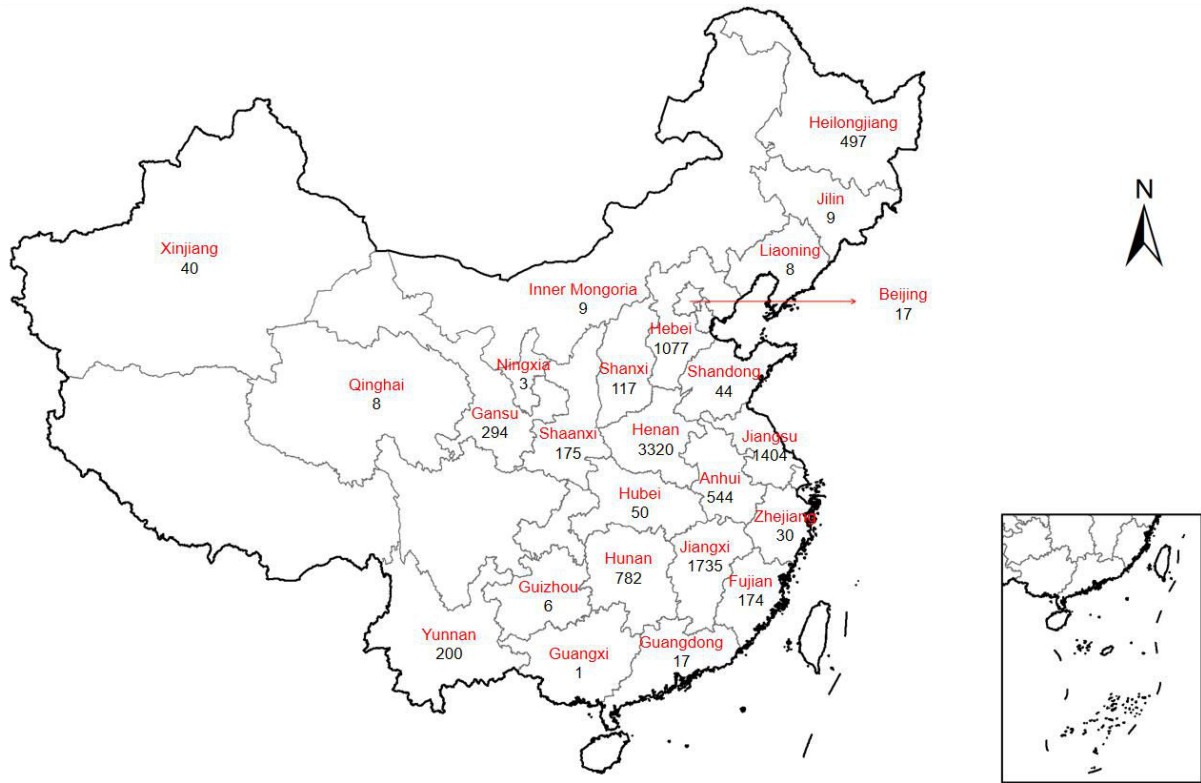

**FIG 1** Distribution of the 10,561 cases by province. ArcGIS 10.7 (ESRI, Redlands, CA, USA) was used to visualize the geographical distribution.

diagnoses, and discharge status, were meticulously extracted from each patient's medical record.

## Statistical analysis

Continuous variables were summarized as means, while categorical variables were expressed as both frequencies and corresponding percentages. Statistical analyses were performed using JMP Pro 17.0 software, and graphical representations were created with Microsoft Excel 16.30 and GraphPad Prism 9.0.

## RESULTS

### Hospital and case distribution

This study included a total of 10,561 cases diagnosed with "pertussis-like syndrome," collected from 33 research centers across 25 provinces in China. The geographical distribution of cases across provinces is illustrated in Fig. 1, while the hospital-specific case distribution is detailed in Table S1.

### Demographic and clinical characteristics

We conducted a comprehensive review and analysis of the demographic and clinical characteristics of the 10,561 cases diagnosed with "pertussis-like syndrome." The baseline characteristics are summarized in Table 1. The majority of patients (69.73%, 7,364/10,561) were infants under 1 year of age. Among the 271 cases admitted to the intensive care unit (ICU), 89.30% (242/271) were infants younger than 1 year, and 83.03% (225/271) were under 6 months of age. Notably, all four recorded fatalities occurred in infants under 1 year, with 75.00% (3/4) being younger than 3 months. A gender disparity was observed among hospitalized children, with boys constituting 56.67% of cases compared

TABLE 1 Basic information of the 10,561 cases

| Variable | No. of cases | Proportion (%) |
|---|---|---|
| Age | | |
| 0–<1 yr | 7,364 | 69.73 |
| 0–<3 mo | 2,918 | 27.63 |
| 3–<6 mo | 3,028 | 28.67 |
| 6 mo–<1 yr | 1,418 | 13.43 |
| 1–<3 yrs | 1,070 | 10.13 |
| 3–<6 yrs | 1,201 | 11.37 |
| 6–<12 yrs | 898 | 8.50 |
| ≥12 yrs | 28 | 0.27 |
| Gender | | |
| Male | 5,985 | 56.67 |
| Female | 4,576 | 43.33 |
| Ethnicity | | |
| Han | 10,363 | 98.13 |
| Non-Han | 198 | 1.87 |
| Residence | | |
| Urban | 5,035 | 47.67 |
| Rural | 5,521 | 52.28 |
| Unknown | 5 | 0.05 |
| Region | | |
| Northeast | 514 | 4.87 |
| North | 1,220 | 11.55 |
| East | 3,931 | 37.22 |
| South | 18 | 0.17 |
| Central | 4,152 | 39.32 |
| Northwest | 520 | 4.92 |
| Southwest | 206 | 1.95 |
| Year of hospitalization | | |
| 2016 | 820 | 7.77 |
| 2017 | 1,134 | 10.74 |
| 2018 | 470 | 4.45 |
| 2019 | 1,630 | 15.43 |
| 2020 | 952 | 9.01 |
| 2021 | 1,439 | 13.63 |
| 2022 | 4,116 | 38.97 |
| Month of hospitalization | | |
| Spring | 2,865 | 27.13 |
| March | 983 | 9.31 |
| April | 935 | 8.85 |
| May | 947 | 8.97 |
| Summer | 2,956 | 27.99 |
| June | 785 | 7.43 |
| July | 963 | 9.12 |
| August | 1,208 | 11.44 |
| Autumn | 2,452 | 23.22 |
| September | 980 | 9.28 |
| October | 800 | 7.58 |
| November | 672 | 6.36 |
| Winter | 2,288 | 21.66 |
| December | 712 | 6.74 |
| January | 820 | 7.76 |
| February | 756 | 7.16 |

**TABLE 1** Basic information of the 10,561 cases (*Continued*)

| Variable | No. of cases | Proportion (%) |
|---|---|---|
| ICU | | |
| Yes | 271 | 2.57 |
| Age distribution[a] | | |
| 0–<1 yr | 242 | 89.30 |
| 0–<3 mo | 147 | 54.24 |
| 3–<6 mo | 78 | 28.78 |
| 6 mo–<1 yr | 17 | 6.27 |
| 1–<3 yrs | 12 | 4.43 |
| 3–<6 yrs | 9 | 3.32 |
| 6–<12 yrs | 7 | 2.58 |
| ≥12 yrs | 1 | 0.37 |
| No | 10,290 | 97.43 |
| Discharge | | |
| Leave the hospital on doctor's order | 9,873 | 93.49 |
| Discharged without a doctor's advice | 537 | 5.08 |
| Transfer to another hospital as directed by the doctor | 12 | 0.11 |
| Death | 4 | 0.04 |
| Other | 135 | 1.28 |

[a]Age distribution of the 271 cases admitted to the ICU.

to 43.33% for girls. Additionally, rural children accounted for a higher proportion of cases (52.28%) compared to their urban counterparts (47.76%).

## Epidemiological characteristics

The temporal distribution of "pertussis-like syndrome" cases, as depicted in Fig. 2, demonstrates a significant escalation in case numbers following the coronavirus disease 2019 (COVID-19) pandemic. These findings reveal a notable upward trend in the incidence of "pertussis-like syndrome" post-pandemic, as illustrated in Fig. 2.

As illustrated in Fig. 3, the annual distribution of "pertussis-like syndrome" cases stratified by age groups reveals notable epidemiological shifts. While infants under 1 year of age consistently represented the majority of cases, accounting for over half of the total

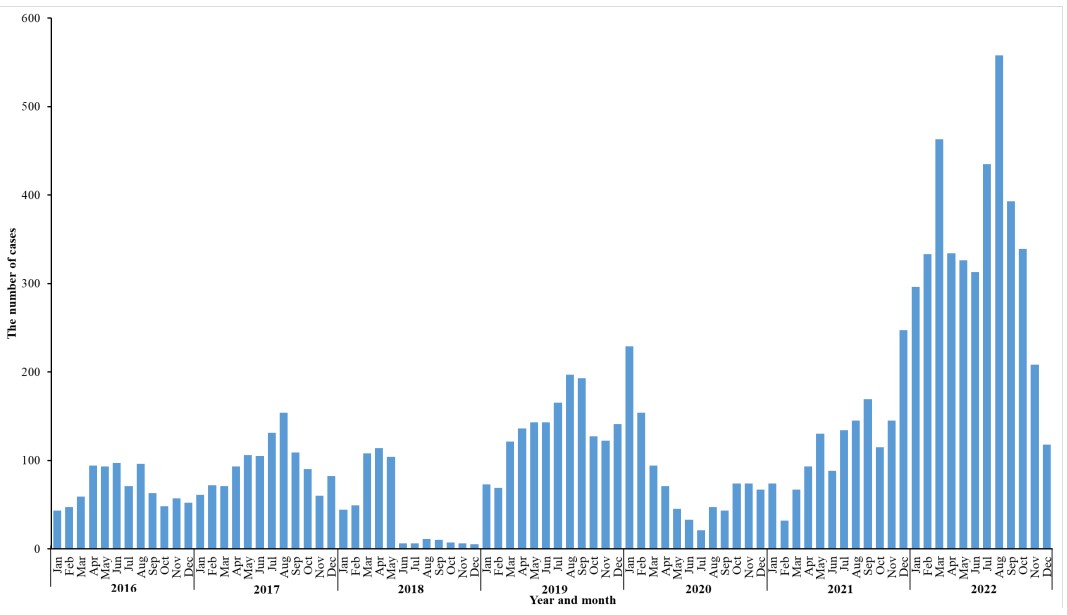

**FIG 2** Monthly distribution of cases diagnosed with "pertussis-like syndrome" in this study.

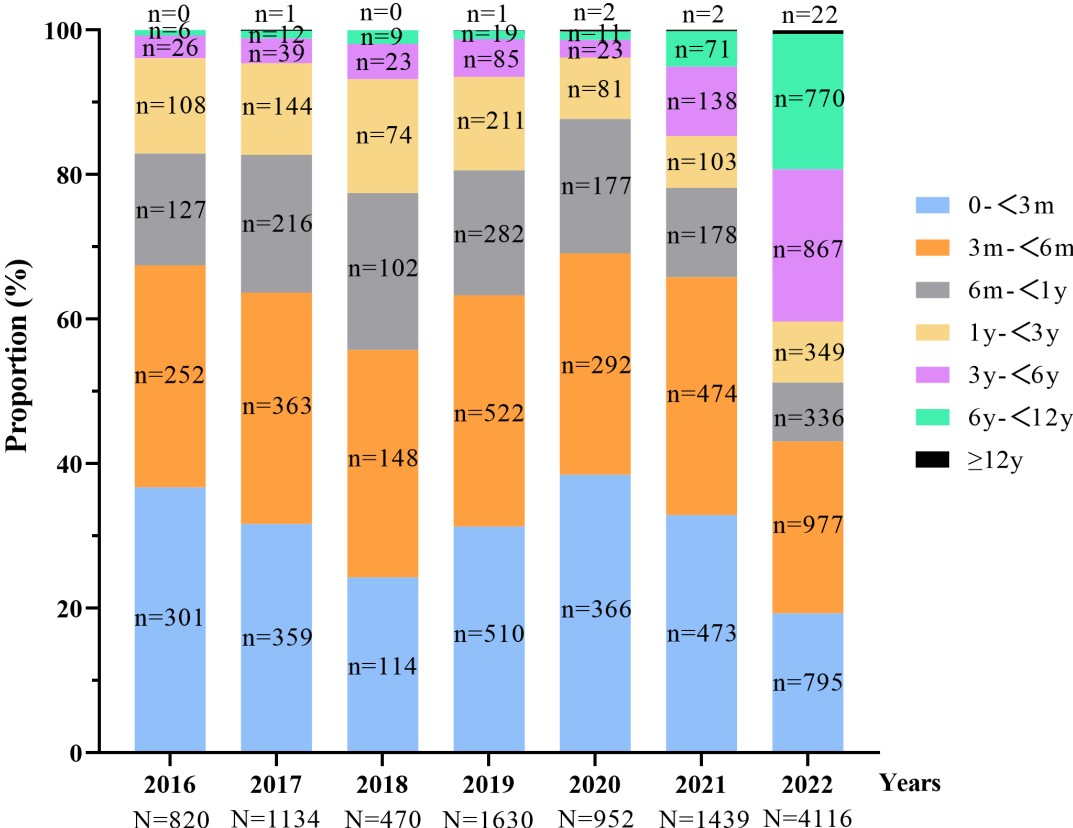

**FIG 3** The proportion of children in different age groups in the annual cases.

cases annually, their proportional contribution exhibited a marked decline from 82.93% in 2016 to 51.21% in 2022. Conversely, the proportion of cases among children over 3 years old demonstrated a significant upward trend, increasing from 3.90% in 2016 to 40.30% in 2022, indicating a changing demographic pattern in disease distribution over the study period.

## Diagnosis and complications

Our analysis of diagnostic and complication profiles demonstrated that a mere 4.37% (462/10,561) of cases were diagnosed with "pertussis-like syndrome" as a standalone condition. The majority of cases presented with concomitant diagnoses, the distribution of which is depicted in Fig. 4.

Lower respiratory infection emerged as the most prevalent complication associated with "pertussis-like syndrome," affecting 93.61% (9,886/10,561) of cases. This was followed by organ dysfunction, which was observed in 11.39% (1,203/10,561) of cases. A detailed breakdown of the 9,886 children diagnosed with lower respiratory infections revealed the following: bronchopneumonia accounted for 4,544 cases, pneumonia (including 844 severe cases) for 3,687 cases, and bronchitis for 1,655 cases.

## Etiological distribution

Out of the 9,886 patients with lower respiratory infections, 1,385 cases (14.01%) had conclusive microbiological findings. Monomicrobial infections predominated, accounting for 79.49% (1,101/1,385) of positive cases, while polymicrobial infections were detected in 20.51% (284/1,385) of cases. Pathogen analysis revealed the following distribution among positive cases: viral agents were identified in 58.77% (814/1,385), bacterial pathogens in 36.25% (502/1,385), *Mycoplasma* species in 25.63% (355/1,385),

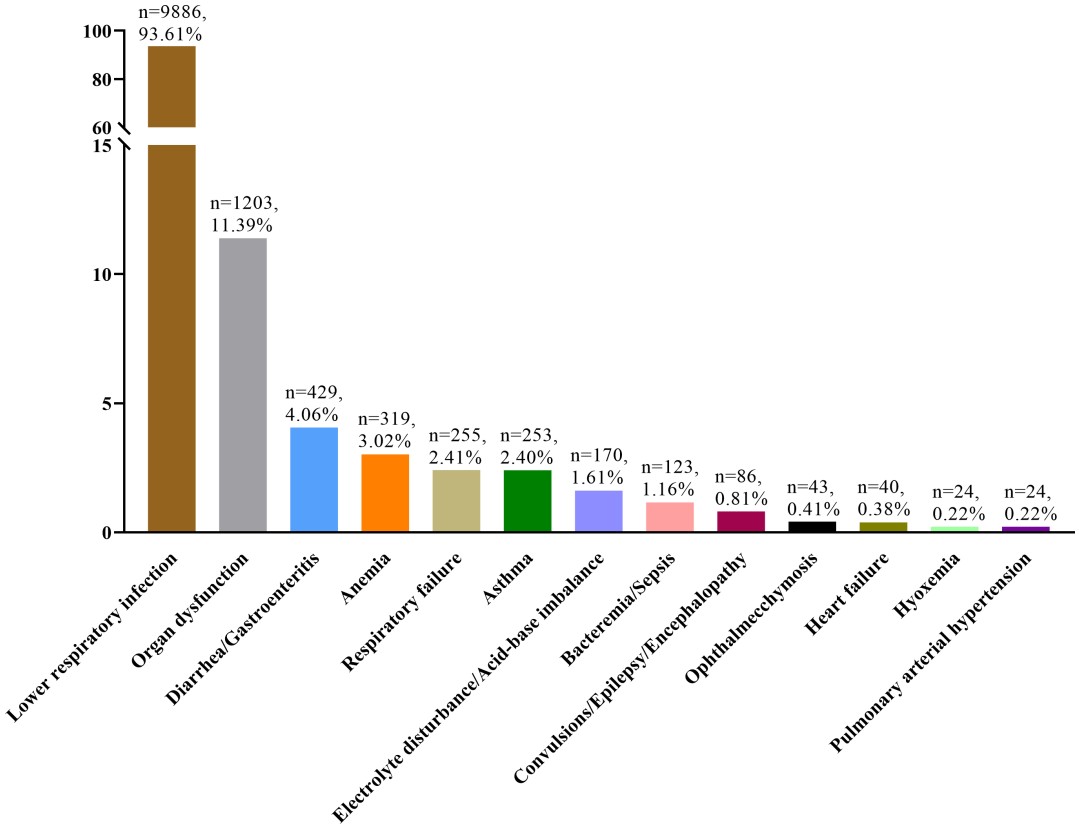

**FIG 4** The distribution of diagnoses other than "pertussis-like syndrome."

and Chlamydia species in 5.85% (81/1,385). Figure 5 presents the comprehensive etiological distribution of pathogens among the 1,385 positive cases.

## DISCUSSION

During the same study period and within the same database, we identified 21,107 cases of pertussis (10) and 10,561 cases of "pertussis-like syndrome," indicating the widespread adoption of "pertussis-like syndrome" in the clinical work across various regions in China. Given its high prevalence, "pertussis-like syndrome" significantly compromises the diagnostic accuracy of true pertussis cases and confounds epidemiological surveillance.

Analysis of the age distribution of children diagnosed with "pertussis-like syndrome" revealed that the majority of cases occurred in infants under 1 year of age. Severe cases requiring intensive care or resulting in death were predominantly observed among infants, particularly those under 6 months of age. This epidemiological pattern mirrors that of pertussis (10), with both diseases predominantly affecting infants under 1 year of age, with younger infants at higher risk of severe outcomes and mortality.

Similar to other respiratory infections (11, 12), the incidence of "pertussis-like syndrome" declined during the COVID-19 pandemic, likely due to non-pharmaceutical interventions such as lockdowns, social distancing, and mask-wearing. However, cases rebounded significantly in the post-pandemic period, mirroring the resurgence of pertussis cases (10).

The age distribution of "pertussis-like syndrome" cases has evolved over time, paralleling trends observed in pertussis (10). While infants under 1 year of age consistently accounted for the majority of cases, their proportion gradually declined over time, while the proportion of cases among older children (over 3 years of age) steadily increased. This shift may reflect the impact of vaccination schedules, heightened awareness of pertussis in older children, and the emergence of more virulent

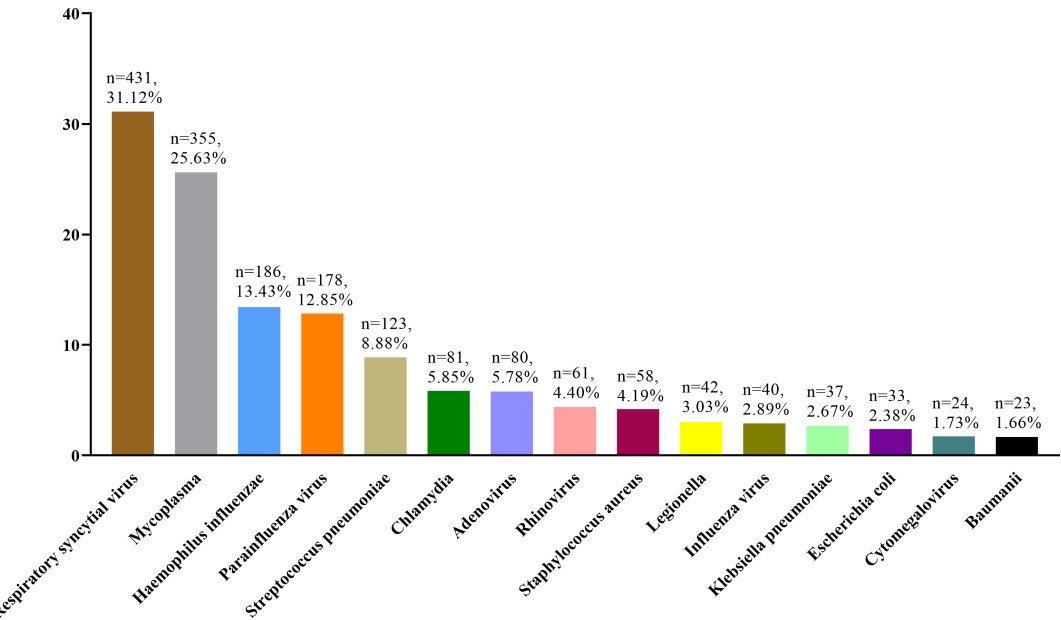

**FIG 5** The etiological distribution of the 1,385 cases with positive results.

or drug-resistant strains of *Bordetella pertussis* (13). The overlapping epidemiological patterns between "pertussis-like syndrome" and pertussis may also reflect the inclusion of undiagnosed pertussis cases within the "pertussis-like syndrome" category.

The diagnosis of "pertussis-like syndrome" highlights the challenges of reliably excluding pertussis through laboratory testing (14, 15). In many healthcare settings, limited access to advanced diagnostic tools often necessitates reliance on clinical criteria for diagnosis. In China, pertussis is classified as a notifiable infectious disease, which requires mandatory reporting to the national infectious disease surveillance system. However, the diagnostic process is complicated by the absence of reliable laboratory confirmation in many instances, resulting in underreporting and a preference for clinical diagnoses such as "pertussis-like syndrome" over confirmed cases of pertussis. Many clinicians refrain from diagnosing pertussis due to limited diagnostic capabilities or to avoid reporting to the national infectious disease surveillance system, choosing instead to initiate direct treatment for the condition.

In this study, the most common comorbidity associated with "pertussis-like syndrome" was lower respiratory tract infection, with respiratory syncytial virus identified as the most prevalent pathogen, followed by *Mycoplasma pneumoniae*, *Haemophilus influenzae*, parainfluenza virus, and *Streptococcus pneumoniae*. These findings are consistent with previous studies on the etiology of "pertussis-like syndrome" (4, 5). Notably, only 4.37% of cases were diagnosed solely as "pertussis-like syndrome," significantly lower than the 9.58% of cases diagnosed solely as pertussis (10). This discrepancy may reflect the broader spectrum of pathogens associated with "pertussis-like syndrome," which includes a wide range of bacterial and viral pathogens, in contrast to pertussis, which is caused exclusively by *Bordetella pertussis*.

As diagnostic methods for identifying common pathogens have become more sensitive and targeted, it is increasingly important to move away from the broad and ambiguous diagnosis of "pertussis-like syndrome" toward more precise pathogen-specific diagnoses. For cases where the causative pathogen remains unidentified, alternative diagnoses such as pneumonia, bronchitis, or bronchopneumonia—conditions listed in the ICD—should be considered. This approach not only facilitates more accurate diagnosis and treatment but also enhances our understanding of the epidemiology and etiology of respiratory infections.

## Conclusions

The diagnosis of "pertussis-like syndrome" is predominantly reliant on clinical manifestations. Its widespread application in clinical practice may lead to an underestimation of the true prevalence of pertussis. Furthermore, the broad spectrum of pathogens implicated in "pertussis-like syndrome" limits its effectiveness in guiding precise therapeutic interventions. In light of significant advancements in pathogen detection technologies, it is advisable to discontinue the use of the "pertussis-like syndrome" diagnosis. Instead, clinical practice should prioritize specific pathogen identification, which would improve diagnostic precision.

## ACKNOWLEDGMENTS

This work was supported by the National Natural Science Foundation of China (81973100) and Funding for Reform and Development of Beijing Municipal Health Commission (EYGF-WSW-04).

The authors sincerely thank the staff members of the Futang Research Center of Pediatric Development (FRCPD) for their assistance and collaboration.

Wei Shi, Xinyu Wang, and Kaihu Yao designed the study and handled the manuscript; Wei Shi and Yahong Hu performed the statistical analyzes; Qinghong Meng, Guoshuang Feng, and Kaihu Yao provided supervision. All authors read, revised, and approved the final manuscript.

## AUTHOR AFFILIATIONS

[1]Beijing Key Laboratory of Core Technologies for the Prevention and Treatment of Emerging Infectious Diseases in Children, Key Laboratory of Major Diseases in Children, Ministry of Education, National Key Discipline of Pediatrics, National Clinical Center for Pediatric Infectious and Allergic Disease Surveillance, National Clinical Research Center for Respiratory Diseases, Laboratory of Infection and Microbiology, Beijing Pediatric Research Institute, Beijing Children's Hospital, Capital Medical University, National Center for Children's Health, Beijing, China
[2]Big Data Center, Beijing Children's Hospital, Capital Medical University, National Center for Children's Health, Beijing, China

## AUTHOR ORCIDs

Wei Shi  http://orcid.org/0000-0001-9269-2312
Xinyu Wang  http://orcid.org/0000-0001-5377-0816
Kaihu Yao  http://orcid.org/0000-0003-1548-8670

## AUTHOR CONTRIBUTIONS

Wei Shi, Conceptualization, Data curation, Formal analysis, Investigation, Methodology, Project administration, Writing – original draft, Writing – review and editing | Yahong Hu, Data curation, Methodology | Qinghong Meng, Supervision | Guoshuang Feng, Supervision | Xinyu Wang, Conceptualization, Writing – original draft, Writing – review and editing | Kaihu Yao, Conceptualization, Writing – original draft, Writing – review and editing

## ADDITIONAL FILES

The following material is available online.

### Supplemental Material

**Table S1 (Spectrum00737-25-s0001.doc).** Distribution of the 10,561 cases in this study.

## Open Peer Review

**PEER REVIEW HISTORY (review-history.pdf).** An accounting of the reviewer comments and feedback.

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
