## [Reviewer comments · Microbiology Spectrum]

Microbiology Spectrum

Identifying the irrationality of the diagnosis of “pertussis-like syndrome” to enhance diagnostic accuracy

Wei Shi, Yahong Hu, Qinghong Meng, Guoshuang Feng, Xinyu Wang, and Kaihu Yao

Corresponding Author(s): Wei Shi, Beijing Children's Hospital Capital Medical University

Review Timeline:

Submission Date:	March 12, 2025
Editorial Decision:	July 21, 2025
Revision Received:	July 24, 2025
Accepted:	August 26, 2025

Editor: Fei Chen

Reviewer(s): Disclosure of reviewer identity is with reference to reviewer comments included in decision letter(s). The following individuals involved in review of your submission have agreed to reveal their identity: Jikui Deng (Reviewer #1)

Transaction Report:

DOI: <https://doi.org/10.1128/spectrum.00737-25>

Re: Spectrum00737-25 (**Identifying the irrationality of the diagnosis of "pertussis-like syndrome" to enhance diagnostic accuracy**)

Dear Dr. Wei Shi:

Thank you for the privilege of reviewing your work. Below you will find my comments, instructions from the Spectrum editorial office, and the reviewer comments.

Revision Guidelines

Sincerely,
Fei Chen
Editor
Microbiology Spectrum

Reviewer #1 (Comments for the Author):

1. In Figure 4, there are 24 cases of pulmonary hypertension, accounting for 0.22%. Have all patients undergone cardiac color Doppler ultrasound?
2. Line 213-214, "Even when pertussis is confirmed, underreporting remains a significant issue." It is inconsistent with the actual clinical situation. Because of the cumbersome reporting process, clinicians often skip laboratory tests when they suspect pertussis, and the actual treatment is still carried out according to pertussis. This happens more often in outpatient clinics.
3. Line 215-220, Although RSV and other pathogens have been detected in the laboratory in these cases of pertussis-like

syndrome, it may still be true pertussis, but there is no specific detection of pertussis pathogens, and these viruses are co-infected.

Reviewer #2 (Comments for the Author):

The manuscript from Shi, et al details the increased prevalence of a diagnosis of "pertussis-like syndrome" (PLS) post-COVID and examines changes in epidemiology before and after 2020.

I found the data in the manuscript to be informative. However, I think an area for the manuscript to improve is in outlining a clear objective. The manuscript concludes that more specific diagnoses would lead to optimizing clinical management. While I think that is likely true, I do not believe the author's provided data to support that fact. The shifting epidemiology with increased frequency and changes of age for diagnosis are clearly demonstrated, but there is no data to suggest that clinical outcomes are impacted for individuals given this diagnosis. There is a push to suggest that there be improve diagnostic testing, but not data supporting that individuals in the group diagnosed with PLS who also had a laboratory finding of RSV or Mycoplasma had improved clinical outcomes. I do think that the manuscript tells a different story that it could focus on in that this epidemiology has shifted and any hypotheses as to why that is. Additional suggestions and points of clarification are below.

1. In the abstract, the resulting section is confusing. I had to read it a few times and didn't really appreciate it until reading the manuscript. The significant increase is mentioned on line 22 but then discusses the decrease in proportion of infants. The language could be a bit clearer.

2. Figure 1 with the distribution of cases should be per population or in another format to help put it into context

3. I had questions about if this was all admitted patients or just all comers including emergency room visits not admitted, that should be more clearly expressed in the results prior to Table 1.

4. For etiological distribution, it was unclear if the 14% success in identification was due to negative tests or what tests were pursued.

5. It was mentioned that some patients with PLS had diagnosis of other infectious agents - is there an explanation of why diagnosis was not updated. Is the diagnosis code maintained based on symptomology or why is there reluctance to update the diagnosis in the face of an alternate diagnosis.

Thanks for the opportunity to review and for sharing your data.

We sincerely appreciate the reviewers for their patience in providing multiple constructive suggestions and granting us the opportunity to revise our manuscript. We have carefully addressed all comments and thoroughly revised the paper accordingly. Below are our point-by-point responses to the reviewers' feedback.

Reviewer #1

1. In Figure 4, there are 24 cases of pulmonary hypertension, accounting for 0.22%. Have all patients undergone cardiac color Doppler ultrasound?

Response: Yes, all patients diagnosed with pulmonary hypertension underwent complete color Doppler echocardiography.

2. Line 213-214, "Even when pertussis is confirmed, underreporting remains a significant issue." It is inconsistent with the actual clinical situation. Because of the cumbersome reporting process, clinicians often skip laboratory tests when they suspect pertussis, and the actual treatment is still carried out according to pertussis. This happens more often in outpatient clinics.

Response: Yes, we could not agree more. In clinical settings, numerous physicians opt for empirical treatment for pertussis without conducting laboratory tests. This practice may be influenced by restricted diagnostic resources in some facilities where clinical testing for pertussis is not routinely performed. However, a considerable proportion of such cases result from a deliberate avoidance of formal diagnosis to circumvent the mandatory reporting obligations within the infectious disease surveillance system. Moreover, many healthcare providers choose not to label laboratory-confirmed pertussis cases as such to avoid reporting requirements. Instead, they categorize them under terms like "pertussis-like syndrome" or similar non-notifiable conditions, contributing to the underreporting of pertussis cases.

We made some slight adjustments to the wording in this section. Page 12, line 204-213.

3. Line 215-220, Although RSV and other pathogens have been detected in the laboratory in these cases of pertussis-like syndrome, it may still be true pertussis, but there is no specific detection of pertussis pathogens, and these viruses are co-infected.

Response: Yes, we fully agree with your perspective. Some cases categorized as "pertussis-like syndrome" with other co-infections may, in fact, be genuine cases of pertussis. These pertussis

cases either did not undergo specific testing for the disease, or if testing was conducted, a diagnosis of pertussis was avoided to evade reporting complications. Hence, we stress the importance of actively conducting pathogen-related tests (including pertussis testing) for infectious cases where the pathogen remains unidentified. In situations where the pathogen is confirmed, it is crucial not to classify them as "pertussis-like syndrome," as this label does not provide meaningful guidance for treatment. Instead, a more definitive diagnosis related to the pathogen should be established. This is the fundamental message we aim to convey throughout our manuscript.

Reviewer #2

The manuscript from Shi, et al details the increased prevalence of a diagnosis of "pertussis-like syndrome" (PLS) post-COVID and examines changes in epidemiology before and after 2020.

I found the data in the manuscript to be informative. However, I think an area for the manuscript to improve is in outlining a clear objective. The manuscript concludes that more specific diagnoses would lead to optimizing clinical management. While I think that is likely true, I do not believe the author's provided data to support that fact. The shifting epidemiology with increased frequency and changes of age for diagnosis are clearly demonstrated, but there is no data to suggest that clinical outcomes are impacted for individuals given this diagnosis. There is a push to suggest that there be improve diagnostic testing, but not data supporting that individuals in the group diagnosed with PLS who also had a laboratory finding of RSV or Mycoplasma had improved clinical outcomes. I do think that the manuscript tells a different story that it could focus on in that this epidemiology has shifted and any hypotheses as to why that is.

Response: Thank you for your insightful comments. The content we presented primarily focuses on the epidemiological characteristics of children currently diagnosed with "pertussis-like syndrome." By comparing the epidemiological features of these children during the same period with published cases of pertussis (Epidemiology of pertussis among pediatric inpatients in mainland China; PMID: 39490385), we discovered numerous similarities between the two groups.

Given the practical issues in the diagnosis and reporting of pertussis, such as some hospitals lacking the clinical testing capabilities necessary for accurate diagnoses or opting not to diagnose

in order to avoid complications with the national infectious disease surveillance system, it is reasonable to speculate that there may be a subset of cases labeled as “pertussis-like syndrome” that are, in fact, true cases of pertussis.

Our analysis of known pathogens in these “pertussis-like syndrome” cases revealed that while a small number had definitive pathogen diagnoses, most lacked clear diagnoses due to insufficient pathogen testing or negative test results. Thus, our primary objective is to encourage clinicians to refrain from diagnosing “pertussis-like syndrome,” as this designation does not provide meaningful guidance for treatment. For patients with available microbiological results, a clear pathogen diagnosis should be established.

In cases with ambiguous pathogen results, intensified efforts should be made to conduct further pathogen testing to achieve more accurate diagnoses and targeted treatments. For cases of pertussis or suspected pertussis, diagnosis should never be evaded. It is imperative to conduct comprehensive etiological testing to achieve a definitive diagnosis. Additionally, diagnosis should never be omitted or deliberately obscured with vague terms such as “pertussis-like syndrome” simply to circumvent mandatory reporting requirements.

We concur with your assessment that our data do not support the conclusion that a clear diagnosis can enhance prognosis or health management. Consequently, we have removed this speculative conclusion, including relevant sections from the abstract and main text, as well as any related content in the research objectives from the original manuscript.

Additional suggestions and points of clarification are below.

1. In the abstract, the resulting section is confusing. I had to read it a few times and didn't really appreciate it until reading the manuscript. The significant increase is mentioned on line 22 but then discusses the decrease in proportion of infants. The language could be a bit clearer.

Response: We apologize for any inconvenience this may have caused and sincerely appreciate your patience in reviewing our document. We have adjusted the wording of this section and divided it into paragraphs to enhance clarity in both logic and expression. Thank you for your understanding. Page 2, line 25-31.

2. Figure 1 with the distribution of cases should be per population or in another format to help put

it into context.

Response: Thank you for your valuable suggestion. Presenting the distribution of cases by demographics would indeed yield more insights into the disease's incidence. However, due to various factors, such as patients residing in different locations and seeking medical treatment elsewhere, our data source is restricted to the residential information provided in the medical records. This limitation makes it challenging to verify the specific circumstances of each case individually. Consequently, Figure 1 currently offers a simplified representation of the geographic distribution of cases.

3. I had questions about if this was all admitted patients or just all comers including emergency room visits not admitted, that should be more clearly expressed in the results prior to Table 1.

Response: We regret not making this point sufficiently clear in our manuscript. The data analyzed in this study were exclusively derived from hospitalized patients and did not include those who visited the emergency department but were not subsequently admitted. To enhance transparency, we added this information to the end of the "Data source" section for the benefit of our readers. Page 5-6, line 103-105.

4. For etiological distribution, it was unclear if the 14% success in identification was due to negative tests or what tests were pursued.

Response: To be precise, the 14% figure does not represent a detection rate; instead, it indicates the proportion of cases with definitive etiological findings among patients diagnosed with lower respiratory tract infections. This study included 10,561 cases sourced from 33 hospitals across 25 provinces. Due to variability in medical resources and testing capabilities among the different institutions, it was not feasible to accurately calculate detection rates for specific tests or individual pathogens. Consequently, we conducted a statistical analysis and presented the composition ratios of each identified pathogen in confirmed cases, which, to some extent, reflects the distribution patterns of these pathogens. We have refined the wording of this section to ensure a more precise expression of its meaning. Page 10, line 165-166.

5. It was mentioned that some patients with PLS had diagnosis of other infectious agents - is there an explanation of why diagnosis was not updated. Is the diagnosis code maintained based on

symptomology or why is there reluctance to update the diagnosis in the face of an alternate diagnosis.

Response: In fact, the cases co-infected with other pathogens were definitively diagnosed etiologically. Our statistical analysis was precisely based on the diagnostic information recorded in each case's medical record front page.

However, for visualization purposes (Figure 4), we classified all lower respiratory tract infection diagnoses, including pathogen-confirmed pneumonia, under the broad category of "lower respiratory tract infection." Subsequently, in Figure 5, we provided detailed stratification by specific pathogen types.

Re: Spectrum00737-25R1 (**Identifying the irrationality of the diagnosis of "pertussis-like syndrome" to enhance diagnostic accuracy**)

Dear Dr. Wei Shi:

Your manuscript has been accepted, and I am forwarding it to the ASM production staff for publication. Your paper will first be checked to make sure all elements meet the technical requirements. ASM staff will contact you if anything needs to be revised before copyediting and production can begin. Otherwise, you will be notified when your proofs are ready to be viewed.

Sincerely,
Fei Chen
Editor
Microbiology Spectrum

Reviewer #1 (Comments for the Author):

This article provides more information to understand the real burden of pertussis in the region.

Reviewer #2 (Comments for the Author):

Thanks for the opportunity to review this revision from Shi, et al on the increased diagnosis of "pertussis-like syndrome". Based on the revisions, as a reader I had a better understanding of the objectives of the work. The authors illustrate the increase in this vague diagnosis and hypothesize that it may be due to limited diagnostic tools or a reluctance for reporting if a conclusive diagnosis is made. I can see how this publication can help illustrate a problem and am interested to see what comes next in terms of getting clinicians to understand and respond to this problem.